

# Extreme wet-cold compound events investigation under climate change in Greece

Iason Markantonis[1,2], Diamando Vlachogiannis[1], Athanasios Sfetsos[1], Ioannis Kioutsioukis[2]

[1]Environmental Research Laboratory, NCSR "Demokritos", 15341 Agia Paraskevi, Greece
[2]University of Patras, Department of Physics, University Campus 26504 Rio, Patras, Greece

*Correspondence to*: Iason Markantonis (jasonm@ipta.demokritos.gr)

**Abstract.** This paper aims to study wet-cold compound events (WCCEs) over Greece for the wet and cold season November-April. WCCEs are divided in two different compound events (TX-RR) and (TN-RR) and two different approaches using fixed (RR over 20 mm/day and Temperature under 0 ºC) and percentile (RR over 95[th] and Temperature under 5[th]) thresholds. Observational data from the Hellenic National Meteorology Service (HNMS) and simulation data from reanalysis and EUROCORDEX models were used in the study for the historical period 1980-2004. Simulation datasets from projection models were employed for the near future period (2025-2049) to study the impact of climate change on the occurrence of WCCEs under RCP 4.5 and 8.5 scenarios. Following data processing and validation of the models, the potential changes in the distribution of WCCEs in the future were investigated based on the projected and historical simulations. WCCEs determined by fixed thresholds were mostly found over high altitudes with a future tendency to reduce particularly under RCP 8.5. On the other hand, WCCEs obtained with percentile thresholds, were distributed mostly in Eastern Greece and Crete while their changes differed significantly among models.

## 1. Introduction

Extreme weather events and their linkage to climate change is a matter of high concern for many scientific groups (Zanocco et al., 2018; Konisky et al., 2016; Curtis et al., 2017). In the last decade numerous scientific researches focused on the causes, the frequency and impacts of extreme compound events (e.g. Aghakouchak et al., 2020; Singh et al., 2021; Sadegh et al., 2018; Zscheischler et al., 2017; Zscheischler and Seneviratne, 2017; Zscheischler et al., 2018). As mentioned in IPCC SREX (Ref 7, p. 118) compound events are defined as: (1) two or more extreme events occurring simultaneously or successively, (2) combinations of extreme events with underlying conditions that amplify the impact of the events, or (3) combination of events that are not themselves extremes but lead to an extreme event or impact when combined. The contributing events can be of similar (clustered multiple events) or different type(s) (Leonard et al., 2014).

The purpose of this article is the study of extreme wet-cold compound events (WCCEs) in Greece during the historical period (1980-2004) and how the occurrence of these events will be affected by climate change. using projection data from and . It has been reported that WCCEs affect the region of Mediterranean Sea, including Greece (Zhang et al., 2021). The examined events belong to the first category of the definition of compound events from IPCC since they refer to the simultaneous exceedance of precipitation and temperature thresholds. WCCEs can have negative impact on people's lives by causing electricity blackouts, affecting agriculture with heavy snowfall or freezing rain, blocking transportation because of closed roads, railways or even airports (Houston et al., 2006; Llasat et al., 2014; Vajda et al., 2014). On the other hand, most of the available freshwater in the country comes from melted mountain snow during spring or summer. Finally, eco-systems, especially on mountains, may be harmed by the absence of snow that climate change may cause (Demiroglu et al., 2015; Pestereva et al., 2012; Trujillo et al., 2012; García-Ruiz et al., 2011). Moreover, Athens, a city of more than 4 million inhabitants, experienced in two consecutive years snowstorms (16, 17 February 2021 and 24 January 2022), which caused great problems in road traffic and electricity failures.. Historically, such events occur infrequently in the region and it is the first time that snow depth exceeds 15cm twice in a period of



eleven months in the city center At other parts of Greece such events are more frequent, and this is shown
in the present study.
This work extends further and more meticulously the study of Markantonis et al., (2021) about daily
minimum temperature and accumulated precipitation WCCEs. The motivation is the absence of such
similar study concerning the country, with few exceptions that used only observational data at some
locations (Lazoglou and Anagnostopoulou, 2019), or modeled data mostly over the broader region of
Mediterranean Sea lacking detailed analysis for Greece (Vogel et al., 2021; Hochman et al., 2021; de
Luca et al., 2020). The greatest part of the study concerns the historical period between 1980 and 2004,
because of the availability of quality controlled daily observational data for minimum temperature (TN),
maximum temperature (TX) and accumulated precipitation (RR). Thence, for that period, we use
observational data from 21 Hellenic National Meteorological Service (HNMS) stations, for the validation
of EURO-CORDEX Regional Climate Models (RCMs), provided by the Climate Change Service of EUs
Copernicus Program and the projection model dataset produced in-house. In addition to the models, two
reanalysis products are included, as the closest to true past climate conditions in regions with no or scarce
observations (Moalafhi et al., 2016). More information about the observational and model datasets is
shown in Section 2. Section 3 highlights the applied methodology while Section 4 presents the
comparison of model data with observations. Section 5, details the results about the WCCEs for the
historical period and the projected changes by each model for the near future period between 2025 and
2049 for two greenhouse gas concentration scenarios, RCP 4.5 and RCP 8.5.
**2. Data**
**2.1 HNMS observations**
HNMS provides freely observational data from 21 stations for the purpose of scientific research. The
data have been formally evaluated by HNMS and the timeseries show no missing or distorted values. In
particular, the timeseries available for the historical period 1980-2004 have a 3-hour temporal resolution
and from these values we have extracted the daily values of TN, TX and RR. Figure 1 shows the position
of the stations while Table A1 of Appendix A provides details on the characteristics of the stations . We
have used the observational data to validate the model datasets with regard to the WCCEs for the
historical period.

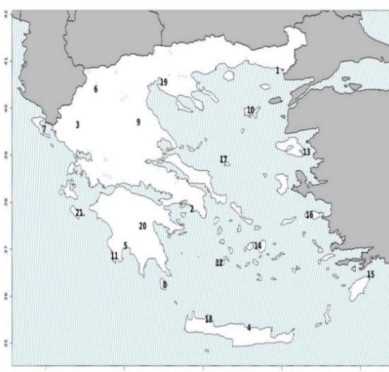

**Figure 1: Map of HNMS stations. The numbers correspond to those in Table A1 (Appendix A).**
**2.2 Reanalysis models**



We have used two reanalysis models due to the lack of spatially and temporally complete direct
observations, to study more consistently the WCCEs in Greece in the historical period. The first model
is the latest available reanalysis product ERA 5 from ECMWF  of spatial resolution ~30km x 30km
(Hersbach et al., 2020). The second reanalysis model, built in Environmental REsearch Laboratory
(EREL) of National Center of Scientific Research 'Demokritos' (NCSRD) WRF_ERA_I, has been
produced by dynamically downscaling ERA-INTERIM using the Weather Research Forecast (WRF)
model (v3.6.1) from 80km x 80km to 5km x 5km (Politi et al., 2021, 2020, 2018).
### 2.3  GCM / RCM models
To observe possible alterations of wet-cold compound events occurrence probability in the future period
2025-2049 compared to the historical period, we employed data from RCM simulations driven by GCMs.
In this regard, we obtained data from 5 models included in the EURO-CORDEX initiative provided by
the Copernicus Program. All chosen models have a spatial resolution of 0.11º x 0.11º and available daily
data for both RCP scenarios. Information on the regional and parent models and their acronyms used
herewith is given in Table 1. In addition to the EURO-CORDEX model data, we have used dynamically
downscaled data from the EC-EARTH GCM to high spatial resolution of 5km x 5km for the area of
Greece using WRF (Politi et al., 2020, 2022)

| Institution | Reference | Regional Model | Forcing model | Acronym | Resolution (°) |
|---|---|---|---|---|---|
| **Météo-France / Centre National de Recherches Météorologiques** | (Spiridonov et al., n.d.) | ALADIN63 | CNRM-CERFACS-CNRM-CM5 | CNRM | 0.11 |
| **Koninklijk Nederlands Meteorologisch Instituut** | (van Meijgaard et al., 2008) | KNMI-RACMO22E | ICHEC-EC-EARTH | KNMI | 0.11 |
| **Climate Limited-Area Modelling Community** | (Rockel et al., 2008) | CLMcom-CLM-CCLM4-8-17 | MOHC-HadGEM2-ES | CLMcom | 0.11 |
| **Swedish Meteorological and Hydrological Institute** | (Samuelsson et al., 2016) | SMHI-RCA4 | MPI-M-MPI-ESM-LR | SMHI | 0.11 |
| **Danish Meteorological Institute** | (Christensen, 2006) | DMI-HIRHAM5 | NCC-NorESM1-M | DMI | 0.11 |
| **EREL (NCSRD)** | (Politi et al. 2020, 2022) | ARW-WRF | EC-EARTH | WRF_EC | 0.05 |


**Table 1: EURO-CORDEX and EREL-NCSRD simulation models information.**

**3   Methodology**
The process we followed in this work is briefly presented in the flowchart of Figure 2. The light blue
steps form the main flow of the approach that mainly include the selection of the compound events based
on threshold criteria, validation of the obtained compounds against observational data, and calculation
of their occurrence probabilities. The models' validation part is a previous step to the exhibition of



modeled data and is added on the data processing step. At the validation step we also compare univariate
20-year return levels using two different approaches, Peaks Over Threshold (POT) and Block Maxima
or Minima (BM), further described in section 4.2.2. The calculated probabilities of WCCEs using all
models in the historical period have been validated against observations. The yellow boxes describe the
results displayed at each step.

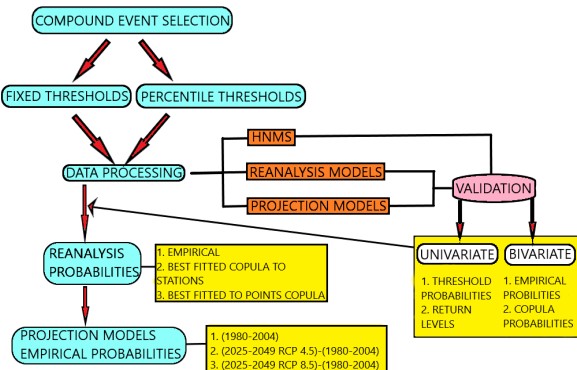


**Figure 2: WCCEs methodology process flowchart.**
In the later sections, we use box-plots to depict the ability of the models to simulate observational data
for the historical period at the cells that include meteorological stations. The box-plots consist of the
colored box, where in the band near the middle of the box is the median, the bottom and top of each color
box are the 25th and 75th percentile (BL) and the ends of the whiskers are the 1.5 times the difference
between the 25th and 75th percentiles (WL).
**3.1 Compound event selection**
According to HNMS the meteorological year can be split into two climate periods
(http://emy.gr/emy/el/climatology/climatology). The cold and wet period extends on average from mid-
October to the end of March, and the warm-dry period occurs during the rest of the year. Since the study
is focused on the extreme WCCEs, we examine the period between November and April, since according
to HNMS observations, April exhibits lower temperatures than October and more rainy days. Moreover,
it is not uncommon for the northern parts of Greece, and especially mountainous areas, to be affected by
snowfalls during April. This leads to the creation of a timeseries of 4532 daily values for the historical
period and 4531 for the future period. The only exception is CLMcom which considers that each month
is consisted by 30 days, thus leading to 4500 values for each period. The near-neighbour approach
revealed the nearest to the station grid cell.
The WCCEs, which are examined on daily basis, are divided in two types of synchronous events, TX-
RR and TN-RR and studied using two different approaches, (1) the percentile threshold and (2) the fixed
threshold (Table 2). For the first method the thresholds are the 95[th] percentile of RR distribution and the
5[th] percentile of TN and TX distribution. This approach examines the threshold for each variable at each
station or grid point. The second approach considers the fixed threshold of 20 mm/day for RR and 0 °C
for TN and TX for all stations or grid points. TN equal to or under 0 °C indicates Frost Days (FD), while
TX equal or under 0 °C Iced Days (ID) (Fonseca et al., 2016). Firstly, we compare the univariate
exceedance probabilities and then the bivariate ones. The difference between the two methods is that the
percentile approach calculates the probability that an event considered extreme for the study area occurs,
while the second that an event considered already extreme occurs. The thresholds examined have been
proposed in various studies for both univariate and bivariate cases (Raziei et al., 2014; Tošić and
Unkašević, 2013; Anagnostopoulou and Tolika, 2012; Pongrácz et al., 2009; Kundzewicz et al., 2006;
Moberg et al., 2006)

| THRESHOLDS | RR | TN | TX | WCCE |
|---|---|---|---|---|




| FIXED | >= 20 mm/day **(RR20)** | <= 0 ºC **(FD)** | <= 0 ºC **(ID)** | 1. **(RR20-FD)** 2. **(RR20-ID)** |
| PERCENTILE | >= 95[th] **(RR95p)** | <= 5[th] **(TN5p)** | <= 5[th] **(TX5p)** | 1. **(RR95p-TN5p)** 2. **(RR95p-TX5p)** |

**Table 2: Univariate thresholds and the compound events examined in the study.**

### 3.2 WCCEs probability calculation

The WCCEs probabilities are calculated applying two different methods. The first is the empirical approach counting the events from the timeseries and dividing by the total number of days to find the percentage (%) of the occurrence probability. For the second method, we use the copula approach for the HNMS observations and models comparison and to map the differences of the two methods for the reanalysis model data. Compared to copula, an empirical method has a higher uncertainty when calculating the probability of extreme events (Hao et al., 2018; Tavakol et al., 2020; Zscheischler and Seneviratne, 2017). The purpose of using two different methods is to examine whether the copula method underestimates or overestimates the WCCEs.

The best fitting copula selection for each timeseries is done using the R programming language function BiCopSelect, included in the package VineCopula (Schepsmeier et al., 2013). The appropriate bivariate copula for each dataset is chosen, by the function, from a multitude of 40 different copula families using the Akaike Information Criterion (AIC) (Akaike, 1974), and the copula chosen for each station and model dataset is shown in Appendix B (Tables B1 and B2). Copulas are used in plenty of studies that investigate the dependence between two different climate variables and the joint probability of compound events (Tavakol et al., 2020; Dzupire et al., 2020; Pandey et al., 2018; Cong and Brady, 2012; Abraj and Hewaarachchi, 2021).

As mentioned in Nelsen, (2007), a bivariate copula is a bivariate distribution function where margins are uniform on the unit interval [0, 1]. A bivariate copula is a map $C:[0,1]^2 \rightarrow [0,1]$ with $C(u,1)=u$ and $C(1,v)=v$. Let X and Y be random variables with a joint distribution function $F(x,y)=Pr(X \leq x, Y \leq y)$ and continuous marginal distribution functions $F_1(x)=Pr(X \leq x)$ and $F_2(y)=Pr(Y \leq y)$, respectively. By Sklar's theorem (Sklar, 1959), one obtains a unique representation

$$F(x,y) = C\{F_1(x),F_2(y)\} \tag{1}$$

For the two random variables of X (e.g., precipitation) and Y (e.g., temperature) with cumulative distribution functions (CDFs) $F_1(x)=Pr(X>=x)$ and $F_2(y)=Pr(Y<=y)$, the bivariate joint distribution function or copula (C) can be written as:

$$F(x,y) = Pr(X>=x, Y<=y) = C(u,v) \tag{2}$$

Besides copula probabilities, we also show the Kendall rank correlation and tail dependence ($\chi$) between the variables (RR–TN) and (RR-TX) to examine the dependence between the variables over all the range and tails of the distribution.

The Kendall rank correlation coefficient evaluates the degree of similarity between two sets of ranks given to a same set of objects (Abdi, 2007) and we prefer it from other correlation types because it provides a distribution free test of independence and a measure of the strength of dependence between two variables. Kendall's tau ($\tau$) is given by Eq. 3, and has a range [-1, 1]:

$$\tau = (Nc–Nd) /(n*(n-1) / 2 \tag{3}$$

where, Nc is the number of concordant pairs and Nd the number of discordant pairs.

Tail dependence describes the limiting proportion that one margin exceeds a certain threshold given that the other margin has already exceeded that threshold that has a range [0, 1]. In R, we use the function taildep from package extRemes (Gilleland and Katz, 2016) for the threshold u=0.95 to calculate Chi ($\chi$). Chi is calculated by:



chi(u) = Pr[Y > G⁻¹(u) | X > F⁻¹(u)] = Pr[V > u | U > u],                                    (4)
where (U, V) = (F(X), G(Y))--i.e., the copula.

## 4    Results in observation locations

In this section, we firstly examine the dependence between the variables based on the HNMS data and
using these data we calculate the probability of WCCEs applying both empirical and copula approaches.
Then, we use the HNMS data to validate both reanalysis and projection models during the historical
period.

### 4.3    HNMS WCCE climatology

Figure 3 presents the tail dependence for the two different types of compound events examined. Only
two stations in Crete show minor dependence between the variables at the tails of the distributions. Figure
4 shows that (RR20-FD) events are located mostly in the mainland, while RR95p-TN5p in the Aegean
Sea area. At several stations, there is a difference between the empirical and the copula approach, which
usually overestimates the total number of WCCEs. In Figure 5a only two stations show a significant
number of RR20-ID events. At the percentile threshold approach (Figure 5b), we observe few WCCEs
using the empirical method, while all stations show a significant number of WCCEs using the copula
method.

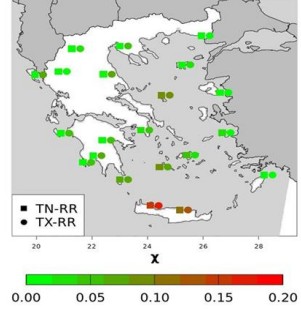


**Figure 3: Tail dependence (χ) for TN-RR (squares) and TX-RR (circles).**

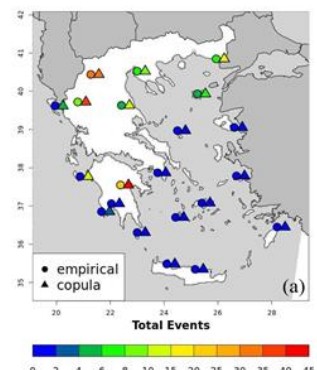 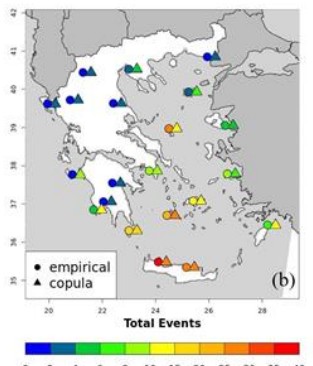

**Figure 4: Total number of WCCEs (1980-2004) for (a) RR20-FD and (b) RR95p-TN5p.**



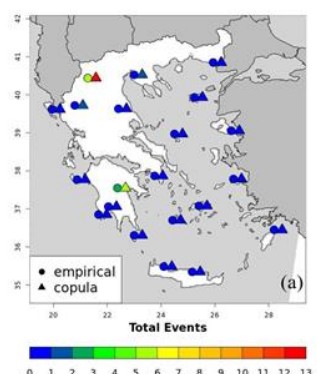 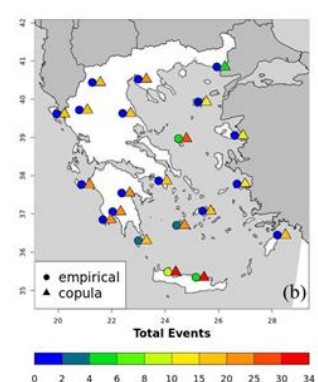

**Figure 5: Total number of WCCEs (1980-2004) for (a) RR20-ID and (b )RR95p-TX5p.**

### 4.4  Univariate validation

Both reanalysis and projections models are compared to observational data for each variable and for the WCCEs probabilities. Figures 6-8 present the mean values and the standard deviation for stations and the respective models' grid points. The corresponding values for each station are shown in Tables S1-S3 and S5-S7 from Supplementary material.

### 4.2.1 Thresholds & Probabilities

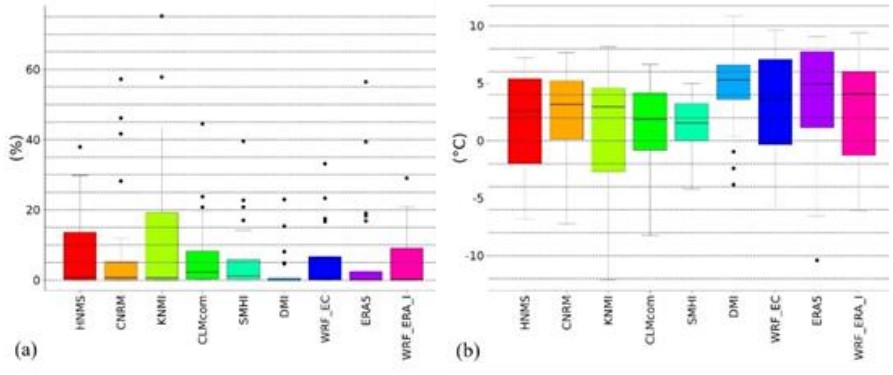

**Figure 6: Boxplots of (a) FD probability and (b)TN5p threshold.**



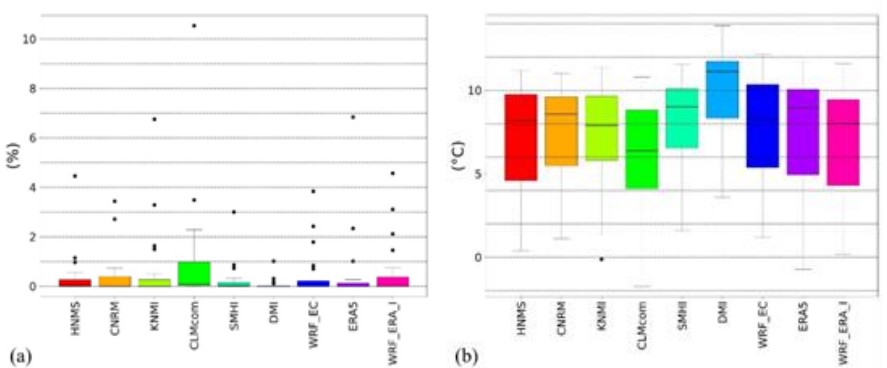

**Figure 7: Boxplots of (a) ID probability and (b)TX5p threshold.**

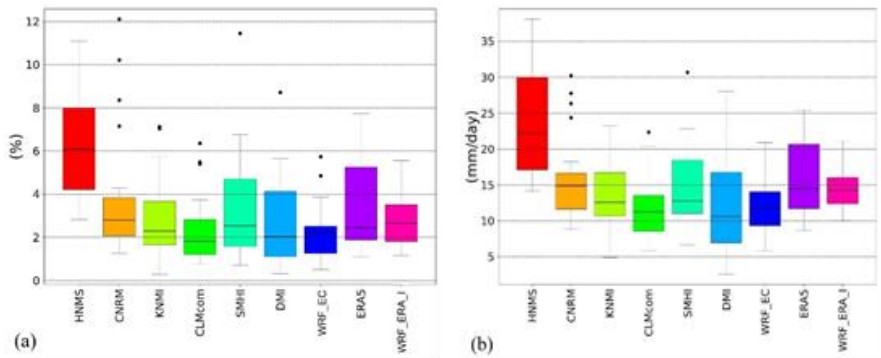

**Figure 8: Boxplots of (a) RR20 probability and (b)RR95p threshold.**

For TN and TX (Figures 6 and 7, respectively) seems to be a good concordance of most models mean values with the HNMS data, although there are differences in the range of BL and WL between the models. The model that mostly overestimates TX5p and TN5p thresholds is DMI. For RR (Figure 8), all models underestimate extreme values compared to HNMS with ERA5 being closer to observations.

### 4.2.2 Return levels

Another way to compare extreme values is the calculation of return levels. As mentioned in methodology we use two approaches, (BM) and (POT). For BM we use the annual maximum or minimum value of the variable that results in the loss of information, because there is available only one value per year. BM samples tend to follow the GEV distribution, according to The Fisher–Tippett–Gnedenko theorem (Fisher and Tippett, 1928; Gnedenko, 1943). For BM we fit the GEV by applying the method 'Lmoments' using the function fevd from R package extRemes.

On the other hand, POT has the advantage of examining more values per year with the chosen condition that the values above the right threshold are considered as extreme (Balkema and Haan, 1974; James Pickands, 1975). The approach is to select as threshold the 90th percentile of the variable distribution (Bommier, 2014). Also, in order to achieve that each extreme value is independent from another, we use a conservative 5-day threshold declustering (Coles, 2001), securing that there are no extreme values affected by the same synoptic system. For POT we fit the Generalized Pareto (GP) distribution, which corresponds to the tail distribution of the GEV (Goda, 2018). As suggested in Poschlod, (2021), we use



Maximum Likelihood Estimation (MLE) as an optimization algorithm to fit the GP to the declustered
timeseries, using again the extRemes package.

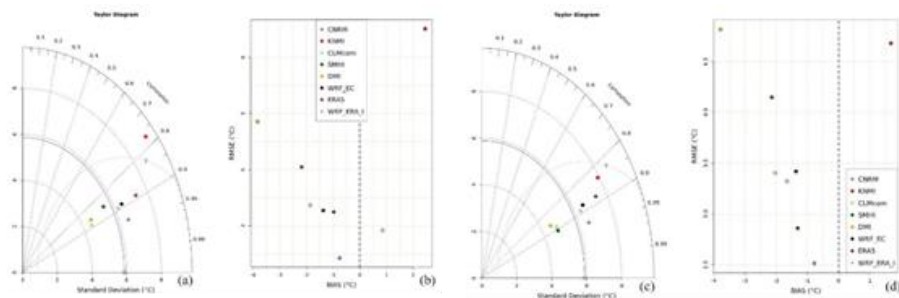

**Figure 9: Taylor diagram for TN 20 years return level using (a) POT and (c) BM approach.**
**RMSE-BIAS plots for (b) POT and (d) BM.**

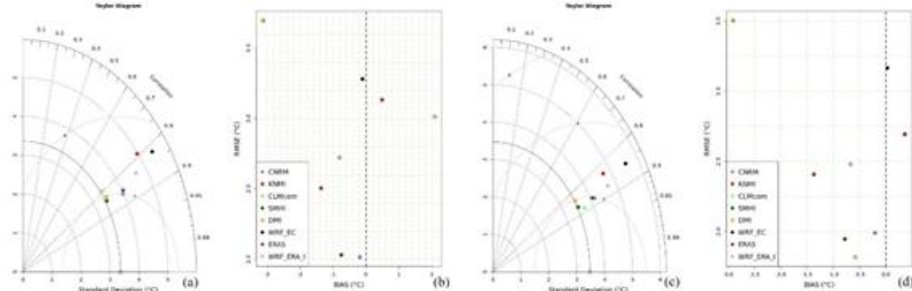

**Figure 10: Taylor diagram for TX 20 years return level using (a) POT and (c) BM approach.**
**RMSE-BIAS plots for (b) POT and (d) BM.**

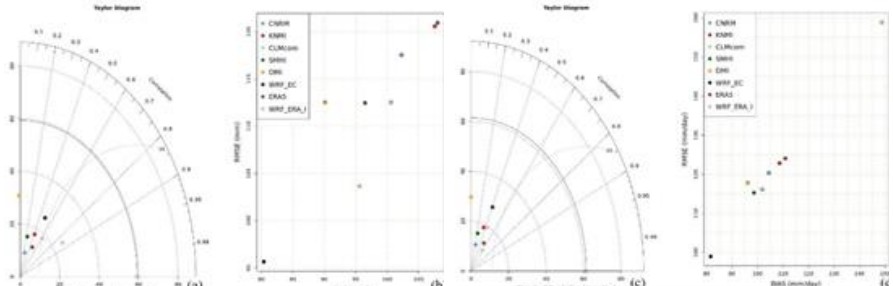

**Figure 11: Taylor diagram for RR 20 years return level using (a) POT and (c) BM approach.**
**RMSE-BIAS plots for (b) POT and (d) BM.**
Figures 9 and 10 show that the CNRM is the model closer to HNMS TN and TX 20 years return level.
Figure 11 yields that WRF_ERA_I has the highest correlation to observations, while WRF_EC the best
RMSE-BIAS relation to observations. The values used to produce Figures 9-11 can be found in Tables
S11-S16 from Supplementary material.

**4.5 Bivariate validation**





The bivariate validation of the models is conducted by the empirical and copula methods for the WCCEs
at the stations. Figures 12 and 13 summarize the results from Supplementary material Tables S4, S5 and
S9, S10, respectively.

258       **4.5.1**    **Empirical approach**


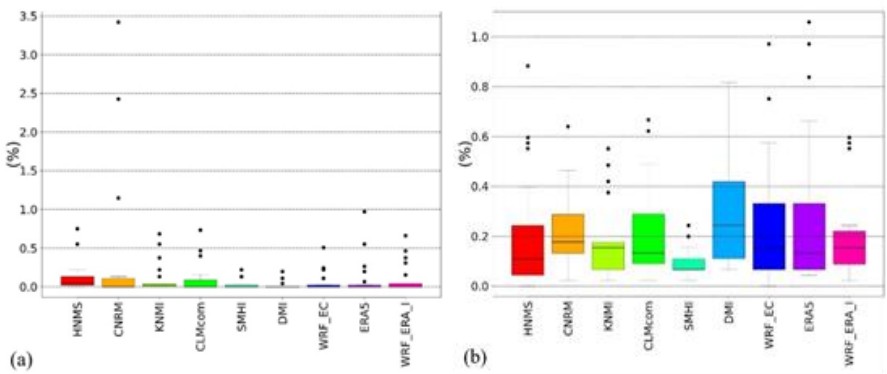

**Figure 12: Boxplots of probabilities for (a) RR20-FD and (b) RR95p-TN5p WCCEs.**

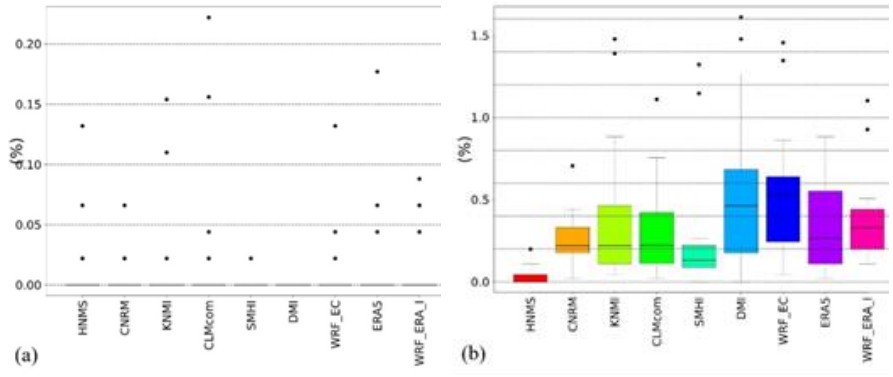

**Figure 13: Boxplots of probabilities for (a) RR20-ID and (b) RR95p-TX5p WCCEs.**
In Figure 12a, HNMS BL is greater than all models, although a number of models show values greater
than the WL of observations, with CNRM yielding the most extreme values, with 3 cases of more than
1% probability. RR95p-TN5p events probabilities from models are close or over the mean values and
BL of HNMS except for the case of SMHI which shows smaller values (Figure 12b). From Figure 13a
we find that RR20-ID events are extremely rare at the locations of the stations with few exceptions. DMI
exhibits zero events, while the largest probabilities are exhibited by CLMcom with four non-zero
probabilities points . In Figure 13b, we see that all models overestimate the probabilities of RR95p-TX5p
events with DMI showing the highest probabilities and SMHI the closer to HNMS agreement.





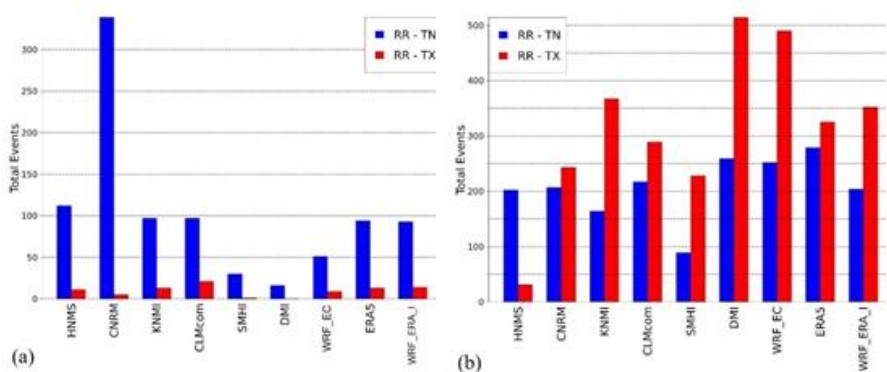

**Figure 14: Bar-plots of total number of WCCEs for (a) fixed and (b) percentile thresholds for the 1980-2004 period.**

In Figure 14, we present a quantitative comparison of the total number of compound events that are counted for all stations and the corresponding grid points for each model. For fixed thresholds, most models show good agreement with the HNMS data except of CNRM which overestimates the amount of total WCCEs for the RR-TN case. Also, SMHI and DMI and to a lesser extent WRF_EC underestimate significantly the number of total events for both types. With the percentile threshold approach all models overestimate the number of WCCEs for the RR-TX case, while for the RR-TN case most models are close to the HNMS total number of WCCEs, except of SMHI which underestimates it.

### 4.5.2    Copula approach

The best-fitted copulas fixed and percentiles probabilities for each model dataset are compared to the respective HNMS station best-fitted copula in Figures 15 and 16, respectively. We use Taylor diagrams and RMSE-BIAS plots to observe which models are closer to the WCCEs probabilities calculated for the HNMS data.

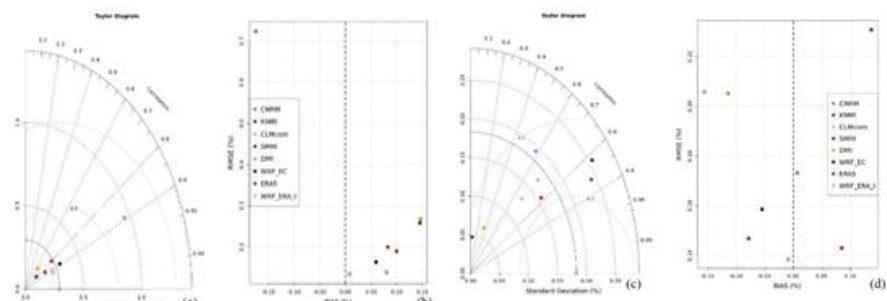

**Figure 15: Taylor diagram of WCCEs copula probabilities for (a) RR20-FD and (c) RR95p-TN5p. RMSE-BIAS plots of WCCEs copula probabilities for (b) RR20-FD and (d) RR95p-TN5p.**




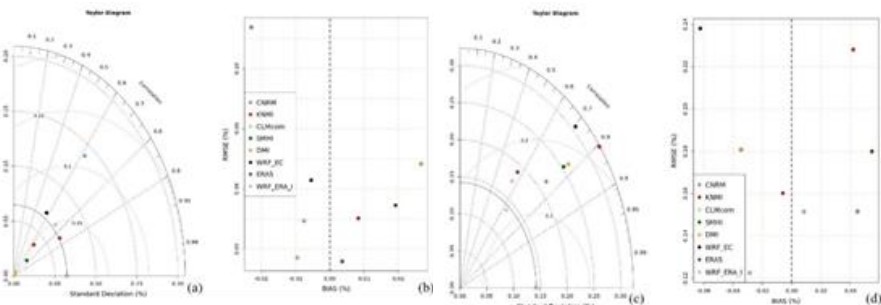

**Figure 16: Taylor diagram of WCCEs copula probabilities for (a) RR20-ID and (c) RR95p-TX5p. RMSE-BIAS plots of WCCEs copula probabilities for (b) RR20-ID and (d) RR95p-TX5p.**

Figures 15 and 16 show that models agree more with observations on fixed thresholds WCCEs than the percentiles ones, where there is a broader deviation of correlation to observations. Probabilities for WCCEs are generally close to zero for observations and models, therefore RMSE and BIAS values are also almost zero. The values for each station are presented analytically in Tables S19-S22 from the Supplementary material.

## 5    Models

### 5.1  Reanalysis

Data from reanalysis models provide us with information on the WCCEs for the historical period, at places with no available observational data. Thus, we will examine the probability of WCCEs using three different methods for the reanalysis data. (1) The empirical probability method, (2) the probability calculated by the most common copula from the total of the 21 HNMS stations and (3) the best-fitted copula at each grid point of the model. For comparison, we show the differences between each pair of methods. The reason to show the second method is to examine its ability to resemble the empirical method, since it is computationally much faster than method (3). In Tables B1 and B2 of Appendix B it is shown that the best fitted copula for HNMS timeseries is the Rotated BB8 270 degrees for (-TN, RR) bivariate distribution and the Survival BB8 for (-TX, RR) bivariate distribution. In both cases, the copulas are chosen for 10 out of the 21 stations. In the appendix, the univariate probabilities and thresholds are also shown. Firstly, we show the Kendall rank correlation ($\tau$) (Figure 17) and then the tail dependence ($\chi$) (Figure 18) between the variables. For the sake of brevity, we refer to the three methods as (A), (B) and (C).

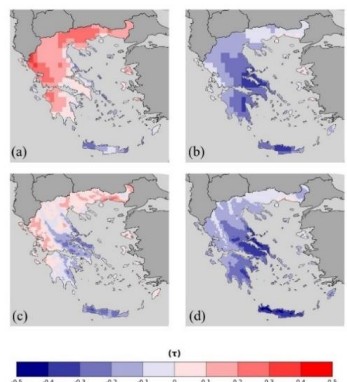

311





**Figure 17: Kendall rank correlation (τ) between (a, c) TN-RR and (b, d) TX-RR and (a, b) ERA 5 and (c, d) WRF_ERA_I.**

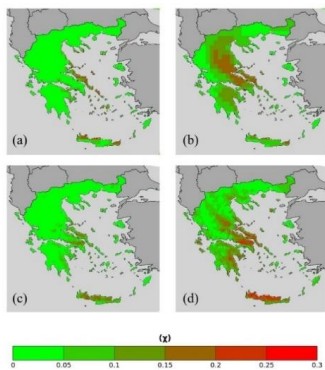

**Figure 18: Tail dependence (χ) at 95% between (a, c) TN-RR and (b, d) TX-RR and (a, b) ERA 5 and (c, d) WRF_ERA_I.**

Figure 17 shows that there is little correlation between the variables with TN-RR having mostly slight positive correlation (17a, 17c), while more negative correlation reaching to -0.5 is calculated for TX-RR (17b, 17d). From tail dependence for the 5 % of the distributions in Figure 18, we see that TX-RR (18a, 18c) are more dependent from TN-RR (18b, 18d) in more regions of the map. Values reach up to 0.3 mainly for TX-RR in eastern Greece and Crete. Also, Figures S1-S3 in th supplementary material present the univariate thresholds and probabilities for RR, TN and TX using the reanalysis datasets (ERA5 and WRF_ERA_I).

### 5.1.1      TN-RR WCCEs

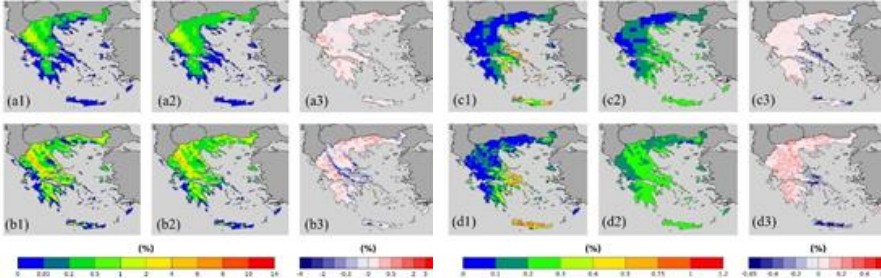

**Figure 19: (a, b) RR20-FD and (c, d) RR95p-TN5p WCCEs probabilities. (a, c) ERA 5 and (b, d) WRF_ERA_I. Column (1) is method A, (2) method B and (3) = (2) – (1).**





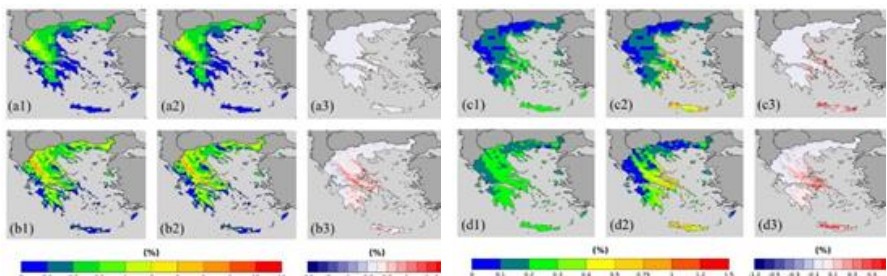

**Figure 20: (a, b) RR20-FD and (c, d) RR95p-TN5p WCCEs probabilities. (a, c) ERA 5 and (b, d) WRF_ERA_I. Column (1) is method B, (2) method C and (3) = (2) – (1).**

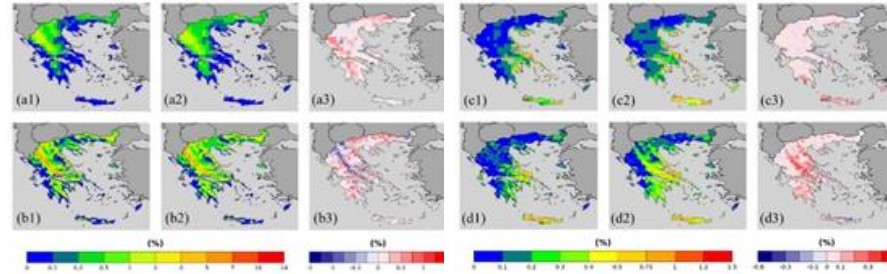

**Figure 21: (a, b) RR20-FD and (c, d) RR95p-TN5p WCCEs probabilities. (a, c) ERA 5 and (b, d) WRF_ERA_I. Column (1) is method A, (2) method C and (3) = (2) – (1).**

From Figures 19 and 20 we observe that method B underestimates the extreme value probabilities compared to methods A and C. On the other hand, method B exhibits less non-zero values compared to method A. In Figure 21, we see that method C mostly overestimates WCCEs compared to method A, especially for RR95p-TN5p and WRF_ERA_I. RR20-FD events reach at most extreme probabilities of 14%, while for RR95p-TN5p the highest probabilities range between 1.2% and 1.5%.

### 5.1.2    TX-RR WCCEs

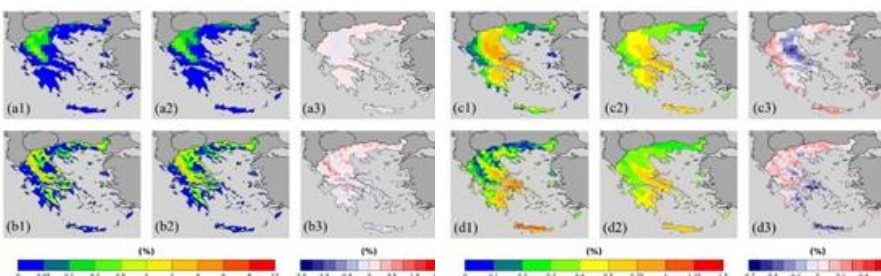

**Figure 22: (a, b) RR20-ID and (c, d) RR95p-TX5p WCCEs probabilities. (a, c) ERA 5 and (b, d) WRF_ERA_I. Column (1) is method A, (2) method B and (3) = (2) – (1).**



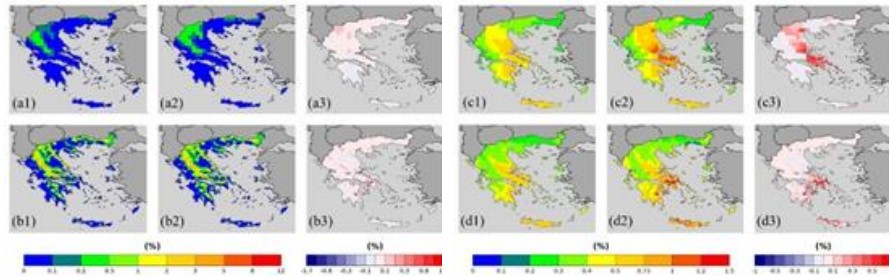

**Figure 23: (a, b) RR20-ID and (c, d) RR95p-TX5p WCCEs probabilities. (a, c) ERA 5 and (b, d) WRF_ERA_I. Column (1) is method B, (2) method C and (3) = (2) – (1).**

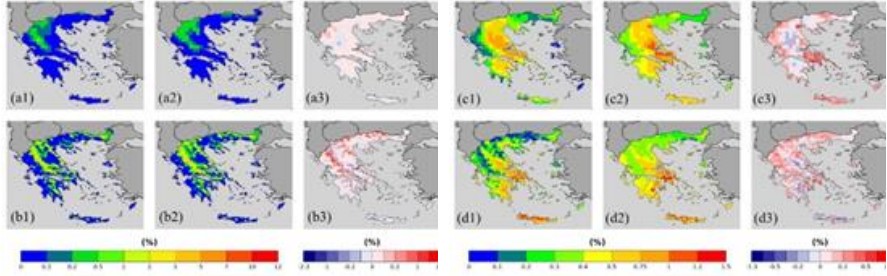

**Figure 24: (a, b) RR20-ID and (c, d) RR95p-TX5p WCCEs probabilities. (a, c) ERA 5 and (b, d) WRF_ERA_I. Column (1) is method A, (2) method C and (3) = (2) – (1).**

Figures 22-24 show that RR20-ID events exhibit lower probabilities than RR20-FD events reaching 10% to 12%. RR95p-TX5p reach 1.5% at the most extreme values, which are distributed at a greater area than RR95p-TN5p. On the other hand, method C exhibits the highest probabilities for both approaches events.

### 5.2 Past-Future Projections comparison

The six projection models we previously evaluated, are used here to study their behavior in the calculation of the probabilities of WCCEs. We compare the historical period probabilities with the probabilities determined for the future scenarios RCP 4.5 and RCP 8.5 for the 2025-2049 period by applying both fixed thresholds and percentiles. The differences mapped are statistically significant at 95% level using the Student's t-test (Goulden, 1939) comparing 25 annual values of the timeseries. We have applied the empirical method to calculate the probabilities of the WCCEs. Univariate thresholds and probabilities are shown in Figures S4-S6 of the supplementary material.





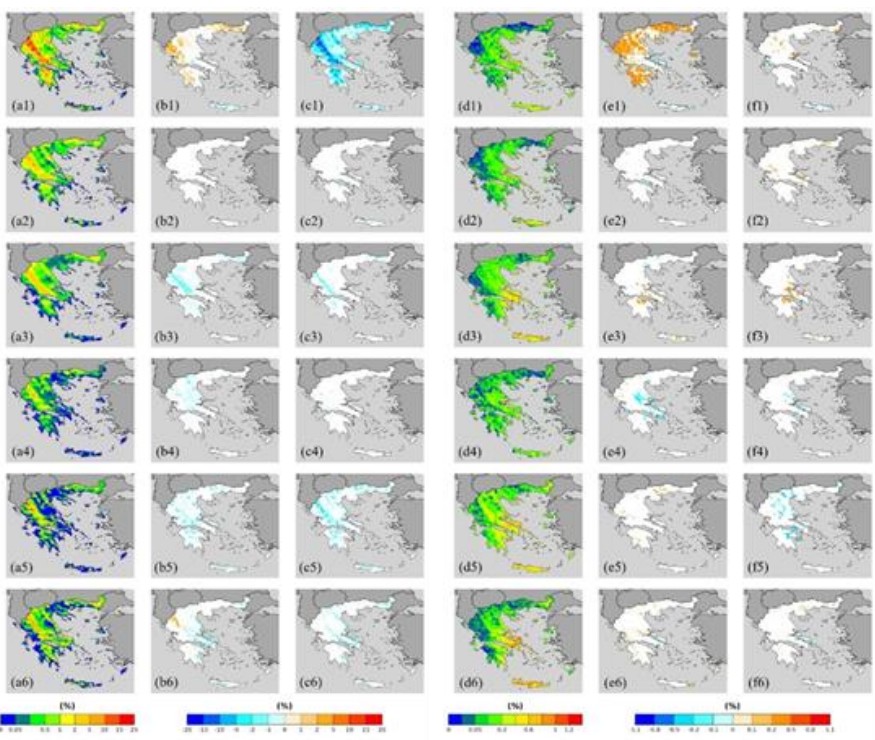

**Figure 25: (a-c) RR20-FD and (d-f) RR95p-TN5p probabilities. Models 1: CNRM, 2: KNMI, 3: CLMcom, 4: SMHI, 5: DMI, 6: WRF_EC. (a, d) 1980-2004, (b, e) (2025-2049 RCP 4.5) – (1980-2004) and (c, f) (2025-2049 RCP 8.5) – (1980-2004).**

We see from Figure 25a that RR20-FD events probabilities may reach 25% particularly for CNRM, which also exhibits the greatest changes in the future, being mostly positive for RCP4.5 and extremely negative (up to -20%) for RCP8.5. Other models calculate fewer extreme probabilities for RR20-FD events and less extreme changes in the future being mostly negative and found in mountainous areas. RR95P-TN5p events displayed in Figure 25d reach up to 1.5% only for WRF_EC. The rest of the models reach most extreme values in the range of 0.4% to 1%. Most models do not display significant changes in the future, except of CNRM which shows positive changes that spread extensively over Greece.



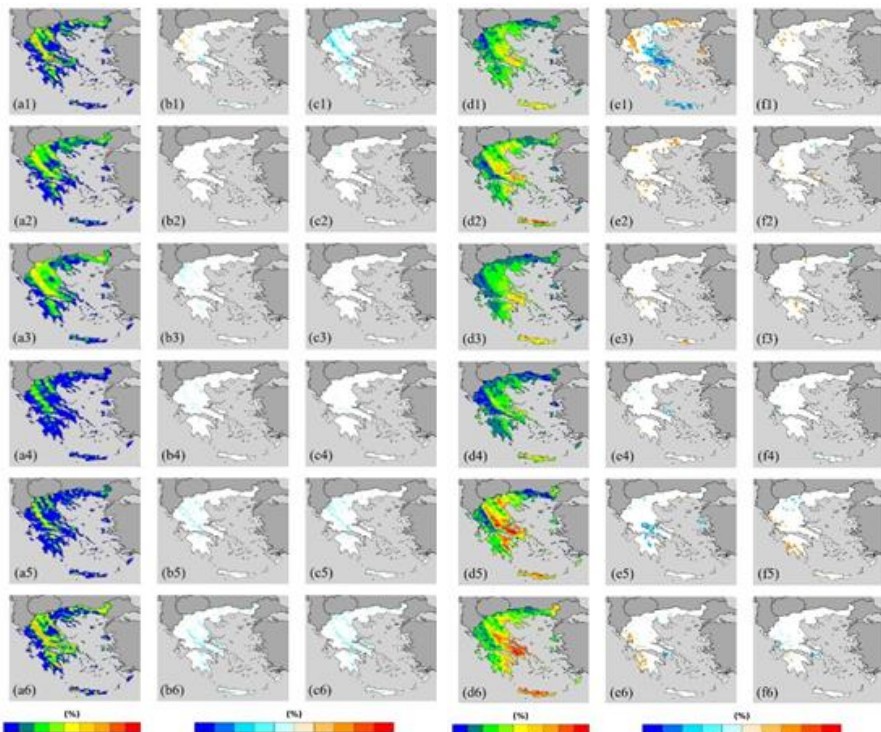

**Figure 26: (a-c) RR20-ID and (d-f) RR95p-TX5p probabilities. Models 1: CNRM, 2: KNMI, 3: CLMcom, 4: SMHI, 5: DMI, 6: WRF_EC. (a, d) 1980-2004, (b, e) (2025-2049 RCP 4.5) – (1980-2004) and (c, f) (2025-2049 RCP 8.5) – (1980-2004).**

Figure 26a shows that RR20-ID events are limited to mountainous areas. Again, CNRM exhibits in few areas the most extreme values ranged between 10% to 20%. Similar values are, also exhibited by WRF_EC. These models display the most extreme reduction of the probabilities in the future, reaching 10% to 15 % in the case of CNRM and RCP8.5. WRF_EC, DMI and to a lesser degree KNMI in Figure 26d, yield the most extreme probabilities for RR95p-TX5p events that reach 1%. The most notable changes are displayed by CNRM under RCP4.5, which shows increases in western and northern parts of the country and significant decreases in eastern areas and Crete.

**Conclusions**

This work presents for the first time to our knowledge an extensive study of wet-cold compound events in Greece for the historical and future periods of 1980-2004 and 2025-2049, respectively. Models' data from EUROCORDEX initiative of 0.11° resolution and reanalysis data (ERA5 and ERA-Interim dynamically downscaled to 5km$^2$) were used and validated for the determined WCCEs against the formally available observational datasets by HNMS for the country. The number of events and their probabilities of occurrence were determined by applying two different approaches, fixed thresholds and percentiles. Then, the validation of the models' datasets against observations was performed for the determined thresholds (univariate and bivariate) and the 20-years return levels using blog-maxima and POT methods. The probability of WCCEs was computed using the empirical method and the best-fitted copula for the bivariate timeseries. Moreover, for the reanalysis data, we applied the approach of the most common copula of the 21 observational stations.

Even though reanalysis and projection models seemed to underestimate extreme precipitation, thus leading to less extreme events, both helped to map the geographical distribution of WCCEs over Greece.



All models agreed that for the historical period, more events by the fixed threshold approach were found
over mountainous regions while the percentile approach yielded more WCCEs over the eastern parts of
the country and Crete.
Furthermore, the projected changes in the number of WCCEs were investigated under RCP 4.5 and RCP
8.5. Significant changes were obtained using the fixed threshold method over mountainous areas which
showed a potential reduction of the number of compound events particularly under RCP 8.5. The
application of the percentile method yielded reduced changes in the probabilities of wet-cold compounds
than the fixed threshold approach while the models showcased higher disagreement among them
concerning the changes.
The reduction of RR20-FD and RR20-ID WCCEs on mountains that most models predicted for the
future, might mean less heavy snowfall events and possibly less accumulated snow depth. If such a
scenario will be verified, Greece faces the threat of losing main sources of fresh water that come from
melted mountain snow during spring or early summer. The change of WCCEs for RR95p-(TN5p or
TX5p) does not necessarily translate to a corresponding change of snowfall events, since the temperature
percentile thresholds are for several occasions higher than 0 °C. Snow events may occur at higher
temperatures, however in this study we examined the amount of precipitation and not its type. Next future
steps could focus on the investigation of the synoptic systems that cause wet-cold compound events in
the area of interest. The higher resolution reanalysis and projection simulations used in the study,
WRF_ERA_I and WRF_EC, exhibited with greater detail the distribution of WCCEs, highlighting the
need for high resolution model data for areas with diverse topography like Greece.

**Acknowledgments**
The authors acknowledge partial funding by the project "National Research Network for Climate Change
and its Impacts, (CLIMPACT - 105658/17-10-2019)" of the Ministry of Development, GSRT, Program
of Public Investment, 2019.

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

**Code and data availability**
Code and results data available upon request.
**Author contributions**
IM has worked on conceptualization, methodology, validation, visualization, investigation, writing
review and editing. AS, DV and IK contributed on conceptualization, review and supervision. All authors
have read and agreed to the published version of the manuscript.
**Competing interests**
The authors declare that they have no conflict of interest.

**Appendix A**

| NUMBER | LOCATION | ID | LATITUDE | LONGITUDE | ELEVATION (m) |
|---|---|---|---|---|---|
| 1 | Alexandroupoli | 16627 | 40.85 | 25.917 | 4 |
| 2 | Elliniko | 16716 | 37.8877 | 23.7333 | 10 |
| 3 | Ioannina | 16642 | 39.7 | 20.817 | 483 |
| 4 | Irakleio | 16754 | 35.339 | 25.174 | 39 |
| 5 | Kalamata | 16726 | 37.067 | 22.017 | 6 |
| 6 | Kastoria | 16614 | 40.45 | 21.28 | 660.95 |
| 7 | Kerkira | 16641 | 39.603 | 19.912 | 1 |
| 8 | Kithira | 16743 | 36.2833 | 23.0167 | 167 |
| 9 | Larisa | 16648 | 39.65 | 22.417 | 73 |
| 10 | Limnos | 16650 | 39.9167 | 25.2333 | 4 |
| 11 | Methoni | 16734 | 36.8333 | 21.7 | 34 |
| 12 | Milos | 16738 | 36.7167 | 24.45 | 183 |
| 13 | Mitilini | 16667 | 39.059 | 26.596 | 4 |
| 14 | Naxos | 16732 | 37.1 | 25.383 | 9 |
| 15 | Rhodes | 16749 | 36.42896 | 28.21661 | 95 |
| 16 | Samos | 16723 | 37.79368 | 26.68199 | 10 |
| 17 | Skyros | 16684 | 38.9676 | 24.4872 | 12 |
| 18 | Souda | 16746 | 35.4833 | 24.1167 | 151 |
| 19 | Thessaloniki | 16622 | 40.517 | 22.967 | 2 |
| 20 | Tripoli | 16710 | 37.527 | 22.401 | 651 |
| 21 | Zakinthos | 16719 | 37.751 | 20.887 | 5 |


**Table A1: HNMS stations information.**





## Appendix B

|  | HNMS | CNRM | KNMI | CLMcom | SMHI | DMI | WRF_EC | ERA5 | WRF_ERA_I |
|---|---|---|---|---|---|---|---|---|---|
| Alexandroupoli | Rot. BB8 270 | Rot Tawn type 2 270 | Rot Tawn type 2 270 | Survival BB8 | Rot BB8 90 deg | Rot Tawn type 1 180 | Gaussian | Frank | Rot BB8 270 |
| Elliniko | Frank | Rot. BB8 270 | Rot BB8 90 | Rot Tawn type 1 180 | Rot Tawn type 2 90 | Rot Tawn type 1 180 | Clayton | Rot Gumbel 270 | Clayton |
| Ioannina | Rot. BB8 270 | Rot BB8 90 | Rot BB8 90 | Rot Tawn type 1 270 | Rot BB8 270 | Rot Tawn type 2 90 | Rot Tawn type 1 270 | Rot BB8 270 | Rot Tawn type 1 270 |
| Irakleio | Gaussian | Rot BB8 270 | Rot Joe 270 | Frank | Rot Tawn type 1 270 | Clayton | Gaussian | BB8 | Survival BB8 |
| Kalamata | Gaussian | Rot Tawn type 1 270 | Frank | Survival BB8 | Rot Tawn type 2 90 | Clayton | Rot Tawn type 1 270 | Rot BB8 270 | Rot Tawn type 2 180 |
| Kastoria | Rot BB8 270 | Rot BB8 90 deg | Rot BB8 90 | Survival Joe | T | RotTawn type 1 180 | Rot Tawn type 1 270 | Rot BB8 270 | Rot Tawn type 1 270 |
| Kerkira | Rot BB8 270 | Rot Tawn type 2 270 | Rot BB8 270 | Survival BB8 | Rot BB8 270 | Rot Tawn type 2 90 | Rot Clayton 90 | Gaussian | Rot Clayton 90 |
| Kithira | Survival BB8 | Tawn type 1 | Gaussian | Survival BB8 | Gaussian | Rot Tawn type 1 180 | Gaussian | Frank | Rot Tawn type 2 180 |
| Larisa | Rot BB8 270 | T | Frank | Survival BB8 | T | Rot Tawn type 1 180 | Rot BB8 270 | Rot BB8 270 | RotTawn type 1 270 |
| Limnos | Rot BB8 270 | Rot Tawn type 2 270 | Frank | Survival BB8 | Gaussian | Rot Tawn type 1 180 | Tawn type 1 | BB8 | Rot Clayton 90 |
| Methoni | Rot Tawn type 2 180 | Rot BB8 270 | Rot BB8 270 | Clayton | Gaussian | Rot Tawn type 1 180 | Rot Tawn type 1 270 | Rot Tawn type 1 270 | Clayton |
| Milos | Gaussian | BB8 | Gaussian | Survival BB8 | Gaussian | Rot Tawn type 1 180 | BB8 | BB8 | Gaussian |
| Mitilini | Rot BB8 270 | Rot BB8 270 | Frank | Rot Tawn type 1 180 | Rot BB8 90 | Rot Tawn type 1 180 | Rot Tawn type 1 270 | Frank | Rot BB8 270 |
| Naxos | Survival BB8 | BB8 | Rot Tawn type 2 270 | Survival BB8 | Rot Tawn type 1 270 | Rot Tawn type 1 180 | Gaussian | BB8 | Rot Tawn type 2 180 |
| Rhodes | Rot Tawn type 2 180 | Tawn type 1 | Rot Tawn type 2 180 | Survival BB8 | Gaussian | Rot Tawn type 1 180 | Rot Tawn type 1 270 | Rot Tawn type 2 180 | Rot Tawn type 1 270 |
| Samos | Rot BB8 270 | Rot Clayton 90 | Rot BB8 90 | Rot Tawn type 1 180 | Rot BB8 90 | Rot Tawn type 1 180 | Rot BB8 270 | Rot BB8 270 | Rot Clayton 90 |
| Skyros | Rot Tawn type 2 180 | BB8 | Rot Tawn type 2 270 | Survival BB8 | Rot Tawn type 2 90 | Rot Tawn type 1 180 | BB8 | BB8 | Gaussian |
| Souda | Gaussian | Clayton | Tawn type 1 | Survival BB8 | Rot Tawn type 1 270 | BB7 | Gaussian | BB8 | Survival BB8 |
| Thessaloniki | Rot BB8 270 | Rot Tawn type 1 270 | Frank | Survival BB8 | Rot BB8 90 d | Rot Tawn type 1 180 | Rot Clayton 90 | Rot Joe 270 | Rot Tawn type 1 270 |
| Tripoli | Rot BB8 270 | Rot Tawn type 1 270 | Rot BB8 90 | Survival BB8 | Rot Tawn type 1 270 | Clayton | Rot Tawn type 2 180 | Rot BB8 270 | Clayton |
| Zakinthos | Rot Tawn type 2 90 | Rot BB8 270 | Rot BB8 270 | Survival BB8 | T | Rota Tawn type 1 180 | Rot Tawn type 1 270 | Frank | Rot Tawn type 1 270 |

**Table B1: (-TN, RR) best-fitted Copula for each station timeseries.**

|  | HNMS | CNRM | KNMI | CLMcom | SMHI | DMI | WRF_EC | ERA5 | WRF_ERA_I |
|---|---|---|---|---|---|---|---|---|---|
| Alexandroupoli | Rot Tawn type 1 270 | Rot BB8 270 | Frank | Rot Tawn type 1 180 | Rot Tawn type 1 270 | Rot Tawn type 2 90 | Rot Tawn type 2 180 | Independence | Rota Tawn type 2 180 |
| Elliniko | Survival BB8 | Rot BB8 270 | Rot Clayton 270 | Rot Tawn type 1 180 | Clayton | Rot Tawn type 1 180 | Gaussian | Gaussian | Gaussian |
| Ioannina | Rot Tawn type 2 180 | Rot Tawn type 2 180 | Rot Tawn type 2 180 | Survival BB8 | Rot Tawn type 2 180 | Survival BB8 | Rot Tawn type 2 180 | Frank | Rot Tawn type 2 180 |
| Irakleio | BB8 | Gaussian | Survival BB1 | Frank | T | Gaussian | Gaussian | Gaussian | Frank |
| Kalamata | Survival BB8 | Rot Tawn type 2 180 | Rot Tawn type 2 180 | Survival BB8 | Survival BB8 | Survival BB8 | Rot Tawn type 2 180 | Rot Tawn type 2 180 | Rot Tawn type 2 180 |
| Kastoria | Survival BB8 | Rot Tawn type 2 180 | Rot Tawn type 2 180 | Survival BB8 | Rot Tawn type 2 180 | Survival BB8 | Rot Tawn type 2 180 | Gaussian | Rot Tawn type 2 180 |
| Kerkira | Survival BB8 | T | Gaussian | Survival BB8 | Rot Tawn type 2 180 | Rot Tawn type 2 90 | Rot Tawn type 1 270 | Rot Tawn type 2 180 | Rot Tawn type 2 180 |
| Kithira | Survival BB8 | Tawn type 1 | Clayton | Survival BB8 | Survival BB8 | Survival BB8 | Gaussian | Rot Tawn type 1 180 | Rot Tawn type 2 180 |
| Larisa | Survival BB8 | Survival BB8 | Tawn type 1 | Survival BB8 | Rot Tawn type 2 180 | Survival BB8 | Rot Tawn type 2 180 | BB8 | Rot Tawn type 2 180 |
| Limnos | Rot Tawn type 2 180 | Rot Tawn type 2 180 | Rot Tawn type 2 180 | Survival BB8 | Rot Tawn type 1 270 | Survival BB8 | Gaussian | Tawn type 2 | Tawn type 1 |
| Methoni | Frank | T | Rot Tawn type 2 180 | Survival BB8 | Survival BB8 | Survival BB8 | Rot Tawn type 1 270 | Survival BB8 | Survival BB8 |
| Milos | Survival BB8 | Rot Tawn type 2 180 | Rot Tawn type 1 180 | Survival BB8 | Survival BB8 | Survival BB8 | Gaussian | BB8 | Frank |
| Mitilini | Rot Tawn type 2 180 | Rot BB8 270 | Rot BB8 270 | Survival BB8 | Rot Tawn type 2 180 | Rot Tawn type 1 180 | Rot Tawn type 2 180 | Rot BB8 270 | Rot Tawn type 2 180 |
| Naxos | Survival BB8 | Rot Tawn type 2 270 | Rot Tawn type 1 180 | Survival BB8 | Survival BB8 | Survival BB8 | Gaussian | Tawn type 2 | Rot Tawn type 2 180 |
| Rhodes | Survival BB8 | Tawn type 1 | Rot Tawn type 1 270 | Survival BB8 | Survival BB8 | Rot Tawn type 1 180 | Rot Tawn type 2 180 | Rot Tawn type 2 180 | Rot Tawn type 2 180 |
| Samos | Rot Tawn type 2 180 | Rot Clayton 90 | Rot BB8 270 | Rot Tawn type 2 180 | Rot Tawn type 2 180 | Survival BB8 | Rot Tawn type 2 180 | Rot BB8 270 | Rot Tawn type 2 180 |
| Skyros | Gaussian | Rot Tawn type 2 180 | Tawn type 2 | Survival BB8 | Survival BB8 | Survival BB8 | Gaussian | BB8 | Survival BB8 |
| Souda | Frank | Gaussian | Gaussian | Frank | T | Gaussian | Gaussian | Frank | BB8 |
| Thessaloniki | Gaussian | T | Tawn type 1 | Survival BB8 | Rot Tawn type 2 180 | Survival BB8 | Tawn type 1 | Rot Tawn type 2 180 | Rot Tawn type 2 180 |
| Tripoli | Survival BB8 | Rot Tawn type 2 180 | Survival BB8 | Survival BB8 | Survival BB8 | Survival BB8 | Survival BB8 | Survival BB8 | Survival BB8 |
| Zakinthos | Frank | Rot Tawn type 2 180 | Rot Tawn type 2 180 | Survival BB8 | Survival BB8 | Survival BB8 | Rot Tawn type 2 180 | Frank | Rot Tawn type 2 180 |

**Table B2: (-TX, RR) best-fitted Copula for each station timeseries.**