# Peer review of "Investigation of the extreme wet-cold compound events changes between 2025-2049 and"

_EGUsphere, 2022_

## Author Response (AR1)

*Reviewer 1*

*This manuscript investigated historical and future wet-cold compound events (WCCEs) over Greece with observation data, reanalysis data and EUROCORDEX models. All models agreed that for the historical period, more events by the fixed threshold approach were found over mountainous regions while the percentile approach yielded more WCCEs over the eastern parts of the country and Crete. Furthermore, the projected changes in the number of WCCEs were investigated under RCP 4.5 and RCP 8.5. WCCEs obtained with percentile thresholds, were distributed mostly in Eastern Greece and Crete while their changes differed significantly among models.*

*This manuscript present too many elementary analysis about historical extremes for observations, reanalysis data and RCMs with different methods (24 figures) without giving a unified conclusion. On the other hand, the subject of this manuscript is compound extremes under climate change, but the discussion about changes in future compound extremes are too simple, with only spatial distribution of extremes. In my opinion, the historical results with obs, reanalysis and RCMS present the historical compound extremes and evaluate performance of RCMs in simulating compound extremes. Then changes in future compound extremes will be discussed in detail. I suggest the authors refine historical compound extremes analysis and reduce or combine some figures, give more discussion about future extremes. Additionally, many grammar errors should be corrected. Thus, major revision is needed.*

Answer: The authors would like to thank the reviewer for the valuable feedback to improve the manuscript. The figures about past period are reduced and greater analysis about the future period is provided.

*List of specific (major and minor) comments:*

*1.      Abstract: Abstract mainly introduce data and methods used in this study without presenting the main results and conclusions. TX-RR, TN-RR, RCP should be given the fullname.*

Answer: Results presented briefly, fullnames written in new version.

*2.      P1, L27: Please give full name of IPCC SREX.*

Answer: Full name is provided in revised version.

*3.      P1, L35: "using projection data from and .": Not a complete sentence. Please confirm.*

Answer: Sentence is properly corrected in revised version.

*4.      Introduction: Recently, there are plenty of studies about compound extremes, and please reorganize Introduction to show the most recent progress about compound extremes, especially for the study area.*

Answer: Thank you for the comment. The introduction has been re-organized. The investigation of compound extreme events is focused on wet-cold compound events. The few latest compound event studies referring to wet-cold compound events are discussed in the introduction.

*5.   P2, L66: This study adopted EUROCORDEX under RCP 4.5 and 8.5. Why not considering CORDEX-CMIP6 under SSPs scenarios?*

Answer: We wanted to adopt models with the finest available resolution, because of the complex topography of Greece. EUROCORDEX at 0.11°. Future works of the authors will consider SSPs scenarios and regional downscaled simulations from CIMP6 global models.

*6.   P2, L71: Is there citation about HNMS observations?*

Answer: HNMS does not provide a specific citation, only the site of the national service is available which has been added in the paper.

*7.   Figure 1: The quality of the figure is too poor to see the details and the the fonts are too small. Please revise the figure and other related figure with higher resolutions and larger fonts.*

Answer: *Figure 1 is replaced by a new version with clearer details and fonts.*

*8.   L85: Give full name of ECMWF.*

Answer: Corrected in script.

*9.   L94: Please give the reason that such five models are adopted.*

Answer: This is answered in the manuscript. These models have the finest available spatial resolution, daily resolution for the periods we examine and also adopted and validated in Cardoso et al., 2019.

*10.   Section 3.1, L132-138: It seems that the definitions of TN5p, TX5p, R95p, R20 are same as those in ETCCDI? If so, please cite it.*

Answer: Thank you for the comment ETCCDI has been added in the script. Also now the study focus only in fixed thresholds.

*11.   Section 4: In my opinion, this section mainly showed the historical results with obs, reanalysis and RCMS and evaluate performance of RCMs in simulating compound extremes in order to investigate future compound extremes with RCMs in Section 5. This section presents too many elementary analysis with too many figures, and some figures are mentioned with few words. Please consider combine similar figures, such as Figs 3-5, 6-8, 9-11 and so on. Additionally, I think compound extremes by observations and reanalysis data are used to evaluate the performance of RCMs in simulating extremes, so Section 5.1 should be mentioned together with observation in Section 4, as well as historical extremes by RCMs. And more deep discussion is also needed.*

Answer:  The authors took into consideration this valuable comment and  required changes have been applied in the paper by changing section and the context in sections 4,5 and 6.

*12.      L195: "4.3 HNMS" should be "4.1 HNMS…", L209: "4.4…" should be "4.2…"*

Answer: Section numbers changed accordingly in revised paper.

*13.      The captions of all figure should be given in more detailed description, such as the meaning of black points in Fig. 6, etc.*

Answer: Box-plots black points are discussed in methodology in revised version. Also, more detailed captions for all figures are provided.

*14.      L258: Please confirm section number: "4.5.1 Empirical approach" and L280: "4.5.2 Copula".*

Answer: Section numbers changed in revised version.

*15.      Since the manuscript mainly focused on compound extreme under climate change, changes in future compound extremes should be given in more detailed discussion. In current version, only spatial distribution of future compound extremes is discussed. Consider giving more discussion about changes in future compound extremes, such as their statistics, multi-model ensemble mean as well as their possible mechanisms, and so on.*

Answer: Mechanisms and further statistical analysis of compound extremes changes will be analyzed in future studies. Ensemble mean is added in the paper as suggested and used as the basis for the future changes of wet-cold compound events.

***Reviewer 2***

*This paper focuses on the wet-cold compound events under climate change in Greece using a series of stations observation, reanalysis, and the historical and projection from the EURO-CORDEX. The research topic may be a relevant to the society, however, due to the poor writing and some infidelity in the data used for validation, I feel a major revision is needed, after a great and careful addressing of the following comments:*

Major comments

*The writing of the whole paper is in a poor state, including some error in words, and vague expression including the title. The title is not good since it only uses a vague naming that covers the scope of the study, but failed with specific details, including the experiments, date time, etc. Such as, the reference seems to be investigating projection of compound events future projection, rather than climate change which may mean present and the future. Some*

*topics like "Investigation of the future extreme wet-cold compound events using EURO-CORDEX regional simulations from 2025-2049."*

Answer: The paper has been changed radically as suggested as well as the title. Now the paper focuses only on fixed threshold wet-cold compound events and the analysis of future changes is conducted using the ensemble mean of the EUROCORDEX projection models. Title changed to "Investigation of the extreme wet-cold compound events changes between 2025-2049 and 1980-2004 using regional simulations in Greece", since besides EURO-CORDEX simulations we use also the build in NCSRD WRF EC EARTH simulation.

*The picture used is in low quality. It is hard to see virtually every taylor diagrams (Figs. 19-25) in the manuscript. Other than that, most of the figures are vague to see, poor in quality, which may need reproduction.*

Answer: All figures are corrected to the proper quality in revised version.

*Question the fidelity of using the reanalysis data since Greece is a mountainous region and the authors' conclusions seems to be largely associated with events on the mountains. There is potential of large cooling temperature and excess rainfall bias in the reanalysis data despite of the , the authors may find supplementary data from archives such as the Global summary of the day or month (https://www.ncei.noaa.gov/access/metadata/landing-page/bin/iso?id=gov.noaa.ncdc:C00516) for supplements to that of the reanalysis data for further evaluation, that would gain more fidelity of the study.*

Answer: Unfortunately, the dataset suggested by the reviewer does not contain observational timeseries in Greece. Also, authors could not find another validated dataset with observations for the historical period studied. Authors acknowledge the excess rainfall bias in low values, or days with zero precipitation, in the reanalysis data which however does not affect this study since its scope are extreme values and the upper tail of the precipitation distribution. Also, precipitation uncertainty is reduced during winter period since convective rainfall is rare during cold season over mountains.

*Creativity issue. The current study fails to go one step forward towards higher creativity. It is obvious that the study of the compound events is not uncommon, whatever the means. The authors haven't significantly separated themselves with these studies other than stating the regional uniqueness for this certain compound event examination. However, we need to note that creativity is insufficient just to use similar method and switching to another region. It may be better if the authors can separate themselves with that of the similar studies of other regions to counter this issue. One possibility is stating the uniqueness of the Greece mountainous regions, and h ow this trait affects the extreme compound event.*

Answer: The uniqueness of the Greece case is a combination of complex orography of the country which mostly affects fixed thresholds compound events since higher probabilities are spotted mainly in higher elevations. These events are mostly caused by the usual west-east movement of synoptic systems in these latitudes. However, Greece located in South Balkans can be affected in winter by Arctic air masses reaching Greece from North.

*What is the take-away message? The author may consider elaborating this part of the work, and how the conclusions drawn from the analysis may be applicable or vary to other mountainous regions around the world, such as that of the Tibetan Plateau, Rocky Mountains, the Andes, and the Alps. This may bring a more valuable message to the broader scientific community.*

Answer: Since the focus of the study now concerns only fixed thresholds, we expect that the rise of temperature will affect drastically the occurrence of wet-cold compound events in other regions of the planet too, although each area has its own unique characteristics .

Minor comments

Line 1, the abstract lacks introduction with the compound events and how it is important to understand. One sentence at least should be used to state the importance.

Answer: A sentence added at the beginning of the abstract.

Line 35, "how the occurrence of these events will be affected by climate change. using projection data from and .", there seems to be a dead sentence just here.

Answer: Error corrected in new version.

Line 57, "thence" -> "hence"

Answer: Error corrected in new version.

---

## Author Response (AR2)

Reviewer 2 comments:

The current paper focuses on the wet-cold compound events under climate change in Greece. The authors have shown improvement in terms of the text and figures after revision of the last manuscript. However, the current study still has considerate flaws, and major concern on the fidelity of the conclusion. Thus, further revision and enhanced study is needed before this study can be considered for publication. I thus give major revision at this point.

*Authors: The authors would like to thank the reviewer for the very constructive comments and recommendations. Here follows a point-by-point reply to the reviewer's questions/suggestions.*

Major comments

1.  It would be good if the authors could draw the topography elevation along with the stations in the Figure 1 to depict the terrain variability, which appears to be a unique factor in the regions studied and concerns the conclusions. This should be easily retrieved by HGT data in ERA or WRF_5.
    *Answer: Figures are corrected and drawn according to the reviewer's suggestion.*

2. Creativity issue. The second paragraph in the introduction should add more studies of wet-compound regions over the world, and extensively discuss what they did and how this study is different from the rest. After the discussion, it is commonly followed by a "However,…." to separate the current studies from the rest. I still don't see this kind of discussion in the text, which does not distinguish the current study from the rest.

In terms of regional extreme studies, how would you define the Greece's special traits, and how this would differ it from other hot research regions with complex terrain such as the Himalayas (extreme high elevation interacting with summer monsoon), Andes mountain (extreme high elevation separating the ocean and land), etc (https://www.ipcc.ch/srocc/chapter/chapter-2/). Secondly, how this unique trait contributes to the future change of wet-cold compound events under the climate change.

*Answer: The Mediterranean Area is a climate change hotspot (IPCC WGII Sixth Assessment Report Cross-Chapter Paper 4: Mediterranean Region) and thus Greece is a special case of study. Although large parts of the country are at some elevations, the highest peak reaches 2918m and the range of the highest altitudes varies between 2000 and 2500 meters, so a comparison with extremely high elevation areas (greater than 3500m) adjacent to oceans is not considered appropriate and relevant.*

3. Fidelity issue. For the WRF_5 data as shown in the following studies have shown a considerate underestimate in temperature and considerate overestimate in rainfall over most of the years.

Politi, N., Vlachogiannis, D., Sfetsos, A. et al. High-resolution dynamical downscaling of ERA-Interim temperature and precipitation using WRF model for Greece. Clim Dyn 57, 799–825 (2021). https://doi.org/10.1007/s00382-021-05741-9

N. Politi, P.T. Nastos, A. Sfetsos, D. Vlachogiannis, N.R. Dalezios, Evaluation of the AWR-WRF model configuration at high resolution over the domain of Greece, Atmospheric Research, 208, 2018, Pages 229-245, ISSN 0169-8095, https://doi.org/10.1016/j.atmosres.2017.10.019.

The conclusion the current studies made is mostly based on the mountainous regions, which for models is commonly to be places of considerable excessive cooler temperature and more rainfall bias. This leads to the still questionable fidelity in the study, as the authors only used sparse observations (for Pindus mountain regions in the northern Greece, maybe only 2 stations) for validation. I understand that observation is commonly sparse in the mountain regions, so one option I recommended is GSOD https://www.ncei.noaa.gov/maps/daily/?layers=0001, it is puzzling for the response since I found 50 stations observations around Greece, with some over the Pindus ranges, which may supplement the HNMS observations the authors used. Also, the cross-validation of several reanalysis used in this study would give an idea of the uncertain range of the model's simulation ability. Note that we shall not overstate the model's ability to project the future without validating and constraining its ability by observation, especially in the complex terrain regions. And this should have a clear discussion in the conclusion part as how potential bias may affect the conclusions drawn in this study.

*Answer: As suggested by the reviewer, we have included all the available stations found in the recommended dataset for Greece concerning the time period under investigation. Here follows the list with the stations found and the available observation days compared to the time period examined in the past.*

|   | ID | NAME | YEARS | LAT | LON | HGT | VALID_OBS |
|---|---|---|---|---|---|---|---|
| 1 | 16643 | AKTIO | 1980-2004 | 38.919 | 20.772 | 2 | 4483 |
| 2 | 16682 | ANDRAVIDA | 1980-2004 | 37.920 | 21.293 | 10 | 4510 |
| 3 | 16675 | LAMIA | 1980-2004 | 38.883 | 22.433 | 12 | 4403 |
| 4 | 16718 | ELEFSIS | 1980-2000 | 38.064 | 23.556 | 20 | 3788 |
| 5 | 16689 | PATRAS | 1980-1999 | 38.250 | 21.733 | 2 | 3447 |
| 6 | 16613 | FLORINA | 1980-2002 | 40.78 | 21.43 | 619 | 4045 |
| 7 | 16665 | ANCHIALOS | 1980-2000 | 39.217 | 22.8 | 19 | 3807 |
| 8 | 16699 | TANAGRA | 1980-2000 | 38.317 | 23.533 | 140 | 3807 |
| 9 | 16706 | CHIOS | 1980-2000 | 38.333 | 26.133 | 5 | 3807 |

*From the list above, the only mountainous station with enough observational data is Florina. Of course, all the stations have been included after a careful removal of missing data since*

*HNMS does not have yet homogenized the data timeseries of these stations. Moreover, all of the models as seen by the validation mostly underestimate the extreme precipitation events examined in this study. Overestimation or underestimation of precipitation from WRF is limited by the fact that model orography is simulated in the range of (0,2200) meters compared to the real orography with peaks reaching 2500-3000 meters*

4. Enhanced analysis is needed. It is not enough just to make simple comparison of the model with "observation" and give a projection without tell readers why. While it is easy to attribute the difference between present and future to climate change, the authors did not mention in what mechanism that is responsible. Whether it's thermodynamic (changes in temperature or rainfall) or thermodynamic (changes in circulation), there should be a mechanism difference for the RCP4.5 and RCP8.5 relative to the historical simulations that is responsible for the change in wet-cold compound events. And this should be further analyzed using the simulation data.

*Answer: The Mediterranean Basin as a climate change hotspot* (Ali et al., 2022) *is expected to experience a rise of temperature as seen also in the supplementary material and mentioned in the text. Near-term future changes in atmospheric circulation are difficult to be modeled and there is low confidence in near-term projections of the position and strength of NH storm tracks. Natural variations are larger than the projected impact of GHGs in the near term* (Qin et al., 2013). *We have added in the text studies that connect wet and cold conditions in the Balkans with teleconnection patterns and possible changes in near future that may affect these conditions.*

Minor

1. Figure 8 Please use 1-6 rather than the a1-3 and b1-3, this does not conform to each other, and would easily puzzle the readers.

*Answer: The format changed according to the recommendations.*

2. The format between the figures are also not consistent, for instance, Figure 10 use (a)(b)(c), while Figure 12, 13 uses A, B.

*Answer: The format changed according to the recommendations.*

3. Line 305 Conclusion add number.

*Answer: Number added and chapter changed to Discussion and conclusions.*

4. Line 317-319, the conclusion is too strong, and I would recommend more conservative discussion rather than conclusions without support of more fundamental facts, especially in the complex terrain region.

*Answer: Thank you for the recommendation. The sentence is altered in the text to a less strong conclusion, although, again we refer to the need to trust our results due to reasons explained in the text.*

---

## Author Response (AR3)

Comments to the author:

Please double check and revise the manuscript again. For example, the line 74 shoule be "2.1 HNMS observations".

*Answer: the authors have checked and revised the script according to editors advise.*